# International Understanding among Nursing and Pharmacy Students in Japan

**Shigeo Yamamura [1],\*** , **Eiko Inoue [2], Junko Miyazawa [2], Kayoko Yuyama [2], Tomoko Terajima [1] and Atsushi Mitsumoto [1]**

[1] Faculty of Pharmaceutical Sciences, Josai International University, Gumyo 1, Togane, Chiba 283-8555, Japan; terajima@jiu.ac.jp (T.T.); amitsumo@jiu.ac.jp (A.M.)

[2] Faculty of Nursing, Josai International University, Gumyo 1, Togane, Chiba 283-8555, Japan; einoue@jiu.ac.jp (E.I.); miyazawa@jiu.ac.jp (J.M.); yuyama@jiu.ac.jp (K.Y.)

\* Correspondence: s_yama@jiu.ac.jp; Tel.: +81-483-53-4583

**Abstract:** The purpose of this research is to establish a model for assessing interest in international understanding among nursing and pharmacy students in Japan. The study design was a cross-sectional survey of nursing and pharmacy students in their first to fourth years at Josai International University. The International Understanding Scale (IUS2000), consisting of four domains (respect for human rights, understanding international culture, awareness of world solidarity, and understanding foreign languages) with 27 items, was used. A path analysis and confirmatory factor analysis were used to model international understanding. The model of international understanding of nursing and pharmacy students was established as the second-order four-factor mode. The international understanding of nursing and pharmacy students was mainly composed of respect for human rights and awareness of world solidarity and was less affected by understanding foreign languages. Nursing students in our study had a higher international understanding than pharmacy students. International understanding was considered relevant to students' learning about the importance of interprofessional collaboration as well as their interests in global learning environments for healthcare professionals. The relationship between international understanding and future progress in healthcare performance needs to be studied to show the importance of international understanding education.

**Keywords:** international understanding; nursing students; pharmacy students; professional education; education for international understanding

## 1. Introduction

International understanding (IU) is defined as "an indivisible whole based on the principle of friendly relations between peoples and States having different social and political systems"; moreover, it focuses "on the respect for human rights and fundamental freedoms", according to the recommendation issued by UNESCO [1]. In 2017, the recommendation was reaffirmed at the General Conference of UNESCO [2]. The recommendation included seven significant principles: "understanding and respect for all peoples, their cultures, civilizations, values, and ways of life, including domestic ethnic cultures and cultures of other nations" [1]. IU is considered an integrated concept that includes knowledge, skills, and attitudes constructive for improving international relations and understanding cultural differences from other countries [3].

The WHO reported that it expects a shortage of 12.9 million healthcare workers worldwide by 2035, indicating that not only healthcare professionals but also students who hope to be healthcare professionals must be conscious of the importance of international collaboration that is based on IU [4]. Since the understanding of cultural difference is integral to IU, increased consciousness could lead to the

development of interprofessional disciplines in healthcare professionals, as reported in the literature. From the viewpoint of professional education, there are increasing calls for IU to be a fundamental competence for nursing and pharmacy students. The global competence of nursing students has been reported to be influential to their future career development because IU can improve critical thinking skills necessary for working with patients from diverse backgrounds [5]. The globalization of pharmacy education can be expected to provide not only broader knowledge, skills, and attitudes but also job opportunities for future graduates [6]. There are other reports suggesting that respect for patients and other healthcare professionals with various backgrounds and cultures is essential to improve competent nursing practice [7] and patient-centered pharmaceutical care [8] in the future.

In relation to IU, the code of ethics for nurses in Japan also states that nurses should provide nursing care regardless of nationality, race, ethnicity, religion, and related factors. [9]. The International Council of Nurses (ICN), in their ICN Code of Ethics for nurses, state that factors such as human rights, cultures, customs, and spiritual beliefs of individuals, family, and community should be respected [10]. For pharmacists, the Japanese Code of Ethics for Pharmacists similarly states that pharmacists should not discriminate by race, social status, philosophy, beliefs, religion, or related factors [11].

These concepts and related findings suggest that IU education in both nursing and pharmacy education programs is important not only to learn about other cultures but also to influence students' future work performance and careers. Therefore, it is valuable to evaluate a level of interest in international understanding and factors influencing IU for nursing and pharmacy students. However, there are few reports about a model of assessing the level of interest in IU in nursing and pharmacy students. If we can assess a degree of interest in IU of students, the effect of IU on their performance or future career development would be evaluated and it will help program development of nursing and pharmacy education.

Josai International University (JIU) has both nursing and pharmacy education programs to educate healthcare professionals with a global mindset [12]. In the previous short report, the impact of early study was found to be abroad programs for first-year nursing students and it was found to promote both social skills and cross-cultural understanding [13].

In this study, we used the Construction of International Understanding Scale (IUS2000) [14] to establish a model for assessing interest in IU among nursing and pharmacy students in Japan. Then, the differences in interest in IU among nursing and pharmacy students and changes in the level of interest in IU with school year were discussed. Furthermore, this study investigated the relationship between IU levels and professional education of nursing and pharmacy students.

## 2. Materials and Methods

### 2.1. Design and Sample

We conducted a cross-sectional survey of nursing and pharmacy students in their first to fourth years at JIU. All nursing and pharmacy students were invited to participate in the web-based survey, and they responded between July and December 2018.

Term lengths of nursing education and pharmacy education programs in Japan are 4 years and 6 years, respectively. We used the responses from first- to fourth-year students of both nursing and pharmacy programs to compare their IU.

For the questionnaire in this survey, the International Understanding Scale (IUS2000) was used to evaluate IU, with permission from the IUS authors [14]. The IUS2000 questionnaire was originally developed to evaluate the effect of education on international understanding for most Japanese students. The ISU2000 consists of 4 domains (respect for human rights, understanding international culture, awareness of world solidarity, and understanding foreign languages) with 27 items. Each domain is further divided into two or three subdomains (Appendix A).

Only students who agreed to participate in the survey after a brief explanation of the purpose of the survey were eligible to respond to the questionnaire. The survey was conducted with a web-based

system, and students rated 27 items on a five-point Likert scale: 1 = strongly disagree, 2 = disagree, 3 = neutral, 4 = agree, and 5 = strongly agree.

The survey was approved by the University's ethics committee (approval number: 02P180029).

*2.2. Statistical Analysis*

The mean score and standard deviation for each item were calculated after treatment for reverse scoring, if necessary.

We supposed that the IU of nursing and pharmacy students consists of four factors according to the four domains in IUS2000. Since each domain in IUS2000 is assumed to be a factor as a latent variable, the items that correlated with a factor (domain) were extracted with reference to the loading obtained in the path analysis. If a factor loading (path coefficient) was smaller than 0.60, the items were removed from subsequent analysis because the factor loading for these items had little correlation with a domain. Cronbach's alpha for each factor was calculated to estimate the integration within the items. Analysis was conducted in the framework of structural equation modeling to allow correlation between factors.

After extraction of items in each factor, IU was modeled with factors, and the fit of the model with the data was examined using goodness-of-fit statistics (comparative fit index, CFI; goodness-of-fit index, GFI; adjusted goodness-of-fit index, AGFI; and root mean square error of approximation, RMSEA). When CFI, GFI, and AGFI are greater than 0.90, and RMSEA is less than 0.07, the fit between the model and data is conventionally considered to be good [15]. The factor scores were calculated with the linear combination of the score and factor score weight and then analyzed as a function of gender, school year, and faculty by multiple regression analysis. The difference of the factor scores with gender, faculty, and school year was evaluated with t-test or ANOVA. SPSS version 26 (IBM Japan, Tokyo, Japan), AMOS version 24 (IBM Japan, Tokyo, Japan), and JMP-pro version 15 (SAS Institute, Tokyo, Japan) were used for statistical analysis.

## 3. Results

Table 1 summarizes the background of the students who participated in the survey. The total number of first- to fourth-year nursing and pharmacy students in JIU was 458 and 531, respectively. Thus, the rates of response in pharmacy and nursing students in the total students were 31.0% and 52.0%, respectively. Table 2 shows the mean scores and standard deviation (SD) of responses to each item in the IUS2000 of nursing and pharmacy students.

Table 3 shows the path coefficients from the factors (domain in IUS) to indicator variables in Domain1 to Domain4. Since the items with small path coefficients (<0.60 in this study) from factors were considered less related to the factor, these items were removed from the indicator variables of the factor, and Cronbach's alpha was calculated using the remaining items (Table 3). Since the values of Cronbach's alpha for the factors are all acceptable (>0.70), extraction of items related to factors was considered reasonable.

**Table 1.** Backgrounds of first- to fourth-year nursing and pharmacy students respondents.

|  | Total N = 419 | Nursing Students N = 142 | Pharmacy Students N = 277 |
|---|---|---|---|
| Gender F/M | 282/132 [(1)] | 129/10 [(2)] | 153/122 [(3)] |
| First-year students | 102 | 40 | 62 |
| Second-year students | 106 | 48 | 58 |
| Third-year students | 103 | 33 | 70 |
| Fourth-year students | 108 | 21 | 87 |

[(1)] 5 missing values, [(2)] 3 missing values, [(3)] 2 missing values.

**Table 2.** Mean and standard deviation (SD) of the responses to each item in IUS2000.

|  | Pharmacy Students | | Nursing Students | |
|---|---|---|---|---|
|  | Mean | SD | Mean | SD |
| ITEM_111 | 4.04 | 0.96 | 3.31 | 1.20 |
| ITEM_112 | 4.35 | 0.82 | 3.84 | 1.06 |
| ITEM_113 | 4.21 | 0.87 | 3.63 | 1.15 |
| ITEM_121 | 4.09 | 1.08 | 3.81 | 1.21 |
| ITEM_122 | 4.40 | 0.87 | 4.09 | 1.02 |
| ITEM_123 | 4.11 | 0.95 | 3.97 | 1.06 |
| ITEM_211 | 2.53 | 1.12 | 2.45 | 1.17 |
| ITEM_212 | 2.42 | 1.14 | 2.16 | 1.27 |
| ITEM_213 | 3.59 | 1.08 | 3.41 | 1.25 |
| ITEM_221 | 4.36 | 0.82 | 3.92 | 1.03 |
| ITEM_222 | 2.64 | 1.14 | 3.18 | 1.21 |
| ITEM_223 | 3.51 | 1.20 | 2.80 | 1.35 |
| ITEM_231 | 4.29 | 0.82 | 3.83 | 1.03 |
| ITEM_232 | 4.47 | 0.77 | 4.16 | 0.98 |
| ITEM_233 | 4.32 | 0.79 | 4.05 | 0.94 |
| ITEM_311 | 3.97 | 0.99 | 3.74 | 0.99 |
| ITEM_312 | 3.84 | 1.15 | 3.47 | 1.20 |
| ITEM_313 | 2.96 | 1.25 | 2.75 | 1.19 |
| ITEM_321 | 3.99 | 0.88 | 3.63 | 1.00 |
| ITEM_322 | 3.66 | 0.97 | 3.33 | 0.97 |
| ITEM_323 | 4.16 | 0.85 | 3.71 | 0.92 |
| ITEM_411 | 2.28 | 1.08 | 2.05 | 1.11 |
| ITEM_412 | 2.53 | 1.00 | 2.29 | 1.13 |
| ITEM_413 | 2.86 | 1.07 | 2.65 | 1.22 |
| ITEM_421 | 3.06 | 1.12 | 3.01 | 1.21 |
| ITEM_422 | 2.94 | 1.25 | 3.12 | 1.37 |
| ITEM_423 | 3.81 | 1.05 | 3.35 | 1.18 |

**Table 3.** Path coefficients from domain to item.

| Factor1 (Domain1) | | Factor2 (Domain2) | | Factor3 (Domain3) | | Factor4 (Domain4) | |
|---|---|---|---|---|---|---|---|
| ITEM_111 | 0.743 | ITEM_211 | 0.235 * | ITEM_311 | 0.555 * | ITEM_411 | 0.604 |
| ITEM_112 | 0.832 | ITEM_212 | 0.186 * | ITEM_312 | 0.340 * | ITEM_412 | 0.486 * |
| ITEM_113 | 0.870 | ITEM_213 | 0.416 * | ITEM_313 | 0.399 * | ITEM_413 | 0.155 * |
| ITEM_121 | 0.259* | ITEM_221 | 0.742 | ITEM_321 | 0.844 | ITEM_421 | 0.601 |
| ITEM_122 | 0.345* | ITEM_222 | 0.409 * | ITEM_322 | 0.711 | ITEM_422 | 0.671 |
| ITEM_123 | 0.261* | ITEM_223 | 0.542 * | ITEM_323 | 0.629 | ITEM_423 | 0.675 |
|  |  | ITEM_231 | 0.690 |  |  |  |  |
|  |  | ITEM_232 | 0.705 |  |  |  |  |
|  |  | ITEM_233 | 0.581 * |  |  |  |  |
| Alpha ** | 0.854 | Alpha ** | 0.765 | Alpha ** | 0.769 | Alpha ** | 0.730 * |

\* These items were removed from the analysis. ** Cronbach's alpha after removing items with *.

　　Figure 1 shows the second-order four-factor model of IU of nursing and pharmacy students with path coefficients. All path coefficients and correlation coefficients were statistically significant. Goodness-of-fit statistics satisfied the conventional criteria (GFI = 0.954 AGFI = 0.932, CFI = 0.968, and RMSEA = 0.052). Some researchers suggested that RMSEA should be lower than 0.05; however, this theory is not theoretically exact. The goodness-of-fit between data and a model should be evaluated with other goodness-of-fit statistics, and a model with RMSEA < 0.07 is sometimes considered to be acceptable [15,16].

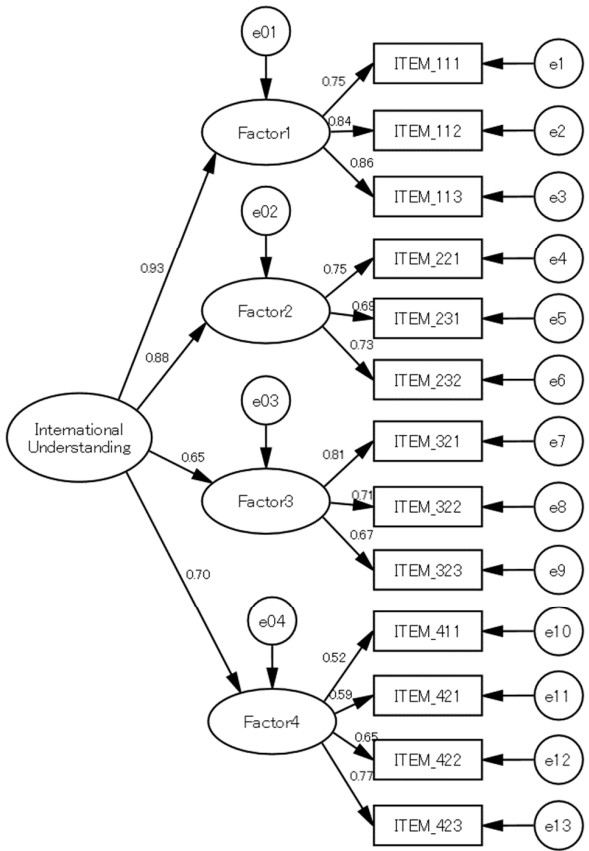

**Figure 1.** Second-order model of international understanding (IU) of nursing and pharmacy students. Goodness-of-fit statistics: GFI = 0.954, AGFI = 0.932, CFI = 0.968, RMSEA = 0.052, AIC = 189.496.

From the model, factor scores of each factor were analyzed as functions of academic department, gender, and school year and are summarized in Table 4. The factor scores of IU and Factors 1 to 4 of nursing students were higher than those of pharmacy students. All factor scores of female students were higher than those of male students. In terms of school year, first - and second-year students had higher IU than senior students.

**Table 4.** Change of factor scores as functions of faculty, gender, and school year.

| Parameter | | International Understanding | | Factor 1 (Domain1) | | Factor 2 (Domain2) | | Factor 3 (Domain3) | | Factor 4 (Domain4) | |
|---|---|---|---|---|---|---|---|---|---|---|---|
| | | Mean (SE) | P | Mean (SE) | P | Mean (SE) | P | Mean (SE) | P | Mean (SE) | P |
| **Faculty *** | Nursing | 4.17 (0.06) | 0.000 | 4.15 (0.07) | 0.000 | 3.84 (0.05) | 0.000 | 4.44 (0.07) | 0.000 | 2.19 (0.04) | 0.001 |
| | Pharmacy | 3.72 (0.04) | | 3.66 (0.05) | | 3.46 (0.04) | | 3.97 (0.05) | | 2.03 (0.03) | |
| **Gender *** | Female | 4.00 (0.04) | 0.000 | 3.95 (0.05) | 0.000 | 3.71 (0.04) | 0.000 | 4.27 (0.05) | 0.000 | 2.15 (0.03) | 0.000 |
| | Male | 3.60 (0.06) | | 3.54 (0.07) | | 3.35 (0.06) | | 3.83 (0.07) | | 1.95 (0.04) | |
| **School Year *** | First | 4.09 (0.07) | 0.000 | 4.05 (0.08) | 0.000 | 3.77 (0.06) | 0.000 | 4.36 (0.08) | 0.000 | 2.24 (0.05) | 0.001 |
| | Second | 4.00 (0.07) | | 3.94 (0.08) | | 3.73 (0.06) | | 4.25 (0.08) | | 2.11 (0.05) | |
| | Third | 3.68 (0.07) | | 3.64 (0.08) | | 3.37 (0.06) | | 3.99 (0.08) | | 1.98 (0.05) | |
| | Fourth | 3.73 (0.07) | | 3.66 (0.08) | | 3.50 (0.06) | | 3.91 (0.08) | | 2.02 (0.05) | |

*, *t*-test, **, ANOVA.

The effect of gender and faculty on IU was evaluated by a regression analysis using categorical variables. There was no interaction between gender and nursing/pharmacy faculty (p = 0.5858), and the results are shown in Table 5. The adjusted IU of female students and nursing students was 0.269 and 0.344 higher than male students and pharmacy students, respectively.

**Table 5.** Effects of gender and faculty on IU.

| Parameter | Estimates | SE | P * |
|---|---|---|---|
| Sex (Female students) | 0.269 | 0.082 | 0.0010 |
| Faculty (Nursing students) | 0.344 | 0.081 | <0.0001 |

* two-way ANOVA. No interaction was found.

## 4. Discussion

The response rates among nursing and pharmacy student participants in this survey were at 31.0% (142/531) and 52.0% (277/531), respectively. Presumably, one reason for this could be because the survey was conducted on a completely voluntary basis. The rates of response were almost the same in each school year, and the factor structure obtained by this survey can be expressed as the IU of nursing and pharmacy students.

All four domains in IUS2000 (respect for human rights, understanding of international culture, awareness of world solidarity, and understanding of foreign languages) were considered as factors of IU of nursing and pharmacy students in JIU. Thus, the second-order model of IU (Figure 1) was well established.

IUS2000 was originally developed with responses from a broad, younger age group of junior high school to university students (102 junior high school students, 727 high school students, and 285 university students) and was evaluated as a good scale for related general knowledge of international affairs of students [14]. ISU2000 was also used to match the background of participants in the survey of English education for high school students [17]. From these results, the IUS2000 with four domains would be more suitable for evaluating IU for younger generations or broader groups [14].

In this research, IUS2000 was also found suitable for assessing the IU of nursing and pharmacy students. The IU of nursing and pharmacy students affected the categories, "respect of human rights" and "understanding international culture" compared with "understanding of foreign language" and "awareness of world solidarity." In the UNESCO recommendation, language is not include in the seven significant principles of IU [1]. Since language is a tool to understand cultural differences, "understanding of foreign language" influences IU less.

Since respect for human rights is a basic competency of education for healthcare professionals, IU of nursing and pharmacy students related to "respect for human rights" is reasonable. Similarly, since nursing and pharmacy students are learning about the importance of interprofessional collaboration with their academic programs, "understanding international culture" would be relevant to their mindset for seeking interpersonal relationships.

The IU of nursing and pharmacy students was modeled based on IUS2000, but not all items were correlated with their IU. Some subdomains or items were less correlated with IU, such as subdomains 12, 21, and 31. The result suggests that the major factor of the IU of nursing and pharmacy students was based on interest in international people with different cultures. There was less interest for their life background, such as inequality between countries or culture (subdomain 12), or their history/religions (subdomain 21) or environmental problems (subdomain 31). Thus, the model of IU of nursing and pharmacy students was similar but not exactly the same in the common younger generation.

As shown in Table 4, factor scores of IU and four factors were significantly different in terms of faculty, gender, and school year categories. The adjusted IU scores of nursing students and female students were higher than those of pharmacy students and male students, respectively (Table 5). For gender differences, female students showed higher factor scores in IU than male students, but no reason was discussed in the report [14]. Female students had higher minority interest than male students [18] and were more willing to interculturally communicate than male students [19]. Interest in minority and willingness to communicate would be related in part to raising IU in female students through interest in other intercultural communication. Since this is just a hypothesis based on previous papers, further discussion of the difference in IU of students between gender is necessary.

The reason for the higher IU in nursing students could be the result of the fact that all nursing students have experienced short-term study in the US. Suzuki et al. suggested that the number of international people who interacted with students had little influence on their IU, but we have found that the experience of studying abroad among nursing students increased their social skills in IU [13,14]. Therefore, the experience of short-term study abroad is suggested as a factor that increases the IU of nursing students.

A tendency to decrease the degree of IU with school year would be considered as this curriculum for nursing and pharmacy students grew more challenging as they moved through the program from basic science or basic skill-oriented subjects to practice- or clinically oriented subjects. So, their learning expectations spread and increased along with the school year. Students may not have had enough time to study global perspectives with their busy curriculum in their third and fourth years. The length of study of pharmacy students was six years, but this survey was carried out for first- to fourth-year students to compare with the IU of the two departmental groups. Pharmacy students experience clinical rotation in their fifth year. Since clinical experience may deepen their perspectives in terms of respect for human rights, the IU of pharmacy students may increase after clinical rotation.

Globalization and international education in nursing and pharmacy education can provide career development or job opportunities [20,21]. IU would be a significant score to assess students' international mindset. For international universities, we need to support students to keep a higher global mindset when they are students.

## 5. Conclusions

We established the second-order IU model for nursing and pharmacy students in Japan including four factors (respect for human rights, understanding of international culture, awareness of world solidarity, and understanding of foreign languages) based on the IUS2000 questionnaire. The IU of nursing students and female students was higher than those of pharmacy students and male students, respectively. This is perhaps because the gender difference for intercultural interest and a short-term early exposure to studying abroad in nursing students would affect the development of IU.

There are some limitations that impact the results of this survey. It is important to note that this survey was conducted only for one group of university students, and participants in the survey were almost half of the total students. This also means that the sample size would not be large enough for a robust analysis. Since generalization of these results may be limited, there is a need to carefully interpret the results of this survey.

In the future, the relationship between IU and future progress in the healthcare performance of students needs to be studied to show the importance of education in international understanding.

**Author Contributions:** Conceptualization, S.Y. and A.M.; statistical analysis, S.Y.; discussion from perspective of nursing education, E.I., J.M., and K.Y.; discussion from perspective of pharmacy education, S.Y., T.T., and A.M. All authors have read and agreed to the published version of the manuscript.

**Funding:** This research received no external funding.

**Acknowledgments:** We acknowledge that the authors who established IUS2000 kindly allowed us to use IUS2000.

**Conflicts of Interest:** The authors have no conflicts of interest directly relevant to the content of this article.

## Appendix A

Questionnaire of ISU2000 (* indicates a reverse code item)

Domain1: Respect for human rights

Subdomain 11 Feelings toward multi-ethnic or multi-nation groups

ITEM_111 I do not want to talk to international people. (*)
ITEM_112 It is fun to get to know people from other countries.

ITEM_113 I want to make friends with many international people.

Subdomain 12: Equality awareness

ITEM_121 It is unavoidable that there is a difference in treatment in each country. (*)
ITEM_122 It is unavoidable that the opinions of people in poor countries were neglected. (*)
ITEM_123 I do not consider that one ethnic group is inferior to other ethnic groups

Domain 2: Understanding international culture

Subdomain 21 Understanding

ITEM_211 I can explain in detail the historical events that occurred in foreign countries.
ITEM_212 I cannot explain the characteristics of the three major religions. (*)
ITEM_213 I know some religions in other countries.

Subdomain 22: Interest

ITEM_221 I want to touch the customs of local people when I go abroad.
ITEM_222 I do not watch TV programs introducing foreign traditional culture.
ITEM_223 I want to know what religions are in the world.

Subdomain 23: Empathy

ITEM_231 I do not want to understand cultures in other countries. (*)
ITEM_232 I think experiencing different cultures is interesting.
ITEM_233 I want to respect the customs of each country.

Domain 3: Awareness of world solidarity

Subdomain 31 Interest in human common issues and recognition

ITEM_311 I want to make efforts to reduce carbon dioxide emissions to prevent global warming.
ITEM_312 I am not interested in maintaining world peace. (*)
ITEM_313 I want to know the state of soil, water, and air pollution caused by waste.

Subdomain 32: Cooperative Attitude to the International Cooperation Agency

ITEM_321 I want to support international organizations working to protect the world's nature.
ITEM_322 I want to support international organizations that are striving to maintain world peace.
ITEM_323 I want to help third world children access to education.

Domain 4: Understanding foreign languages

Subdomain 41: Understanding

ITEM_411 I can read newspapers and magazines written in foreign languages like English.
ITEM_412 I can express what I want to say in a foreign language (i.e., English).
ITEM_413 I cannot answer when asking from a foreigner in English.

Subdomain 42: Interest

ITEM_421 I am not interested in newspapers and magazines written in foreign languages.
ITEM_422 I do not want to take a foreign language exam or certification (TOFLE, TOEIC, etc.) (*)
ITEM_423 I do not want to learn a foreign language. (*)

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
