# Peer review of "International Understanding among Nursing and Pharmacy Students in Japan"

_education, doi:10.3390/educsci10090253_

Round 1

Reviewer 1 Report

This manuscript describes the opinion of nursing and pharmacy students in Japan with respect to various aspects of ‘international understanding’, measured as scores on (subscales of) a 27-item self-report questionnaire (the IUS2000). The study appears to be carried out correctly, but the presentation of data can be improved to enhance readability

  1. Table 2, representing the research data, is unnecessarily large and can be reduced to presenting the means ± SD of the Nursing and Pharmacy students only. Skewness and p-values are not necessary because in the Results section of the manuscript no direct comparisons of item scores are made (the results are analysed only on the basis of factor scores), and no consequence or discussion is attached to the negative skewness values observed (not surprising in ceiling scores on the 5-point Likert scales). If the authors want to be transparent to the research community their original data (individual scores for all questionnaire items and dummy variables indicating gender, study programme and study year) are better submitted to a database (a policy supported by the Journal).
  2. In my opinion Figure 1 is unnecessary. The schematic is illustrative only and well-known, while the relevant path coefficients can be found in Table 3. Moreover, the schematic representation is repeated partly in Figures 2 and 3.
  3. The authors state (lines 165-166) that the second order SEM-model (figure 3) is ‘improved’ compared to the four-factor model (figure 2). I think that this comparison is an incorrect one, because both models are essentially the same (the only difference being that the multiple inter-factor correlations are changed into a new latent factor and four path coefficients). This is confirmed by the fact the most fitting parameters (GFI, AGFI, CFI) are identical and that changes in RMSEA and AIC are only minimal. The really important comparison between models would be between the second-order model (figure 3) and a model where all items are loading directly on one latent factor ‘international understanding’. The result of this comparison could give an answer to the question whether the four domains of the IUS2000 can be considered separate constructs or not.
  4. Related to the previous remark, this reviewer would like to see the results of an exploratory factor analysis (EFA) with oblique rotation. In table 3 path coefficients of four separate confirmatory factor analyses are given and a large number of items were deleted based on path coefficients being smaller than 0.6. Given the high inter-factor correlations (figure 2) I wonder whether any cross-factor loadings between the four factors were observed. A more conventional way of representing the data in Table 3 would be to present the factor loadings of an EFA, with omission of loading < 0.3 or < 0.4. The results of this analysis could also help in answering the question at the end of my previous remark.
  5. Interpretation of the results is influenced strongly by omitting a large number of questionnaire items from the analysis (indicated in Table 3 and the Appendix). It is remarkable that many value-laden items (subdomain 12, item 233 and subdomain 31) and the items related to spoken English (412 and 413) are not included in the final analysis. This is remarked in the Discussion (lines 220-227) but could be discussed more elaborately in the light of the UNESCO recommendations (lines 24-27). By omitting these items from the analysis the validity of the IUS2000 as an instrument to assess all aspects of ‘international understanding’ could be questioned.
  6. Lines 172-175 and Table 5: The comparison between male and female students on one hand, and the comparison between nursing and pharmacy students on the other hand is described in an unclear way. The results are better given as the results of a two-way ANOVA, where gender, faculty and gender-faculty interaction are considered as separate sources of variation. How about including school year in a three-way ANOVA? The results, as mentioned in table 4, suggests this to be an effective interpretation tool.
  7. How about comparison to other countries? Is there any literature available from other countries than Japan which has investigated the (multi)cultural awareness of pharmacy and/or nursing students?

Minor remarks:

  1. Line 98-99: “… to the factor loading obtained by the path analysis …” probably should be “… to the loading obtained in the factor analysis …”. Note that Path analysis is a special form or Structural equation modelling.
  2. Line 103: “path analysis” should be “factor analysis”
  3. Line 109 and many other places in the manuscript: “factor points” should be “factor scores”
  4. Line 128: “path analysis” should be “factor analysis”.
  5. Table 4: Statistical significance of the comparisons between school years probably is tested by an F-test, not a t-test. This should be mentioned in the Methods section and as a footnote in the table.

Author Response

Thank you for all of your detailed comments and suggestions.

We modified our manuscript following your valuable comments and suggestions.

1. Thank you for your suggestions. Actually, table 2 was too large and provided little information to discussion. We modified to delete the columns of skewness and p-values. We can also provide all our data to the Journal.

2, 3. I agree to this your suggestion. We removed Figures 1 and 2. Information in Fig 1 are incorporated into Table 3. Because our purpose of this research is to model the IU of nursing and pharmacy students, we decided to remove figure 2 also. We intended to establish better fitting model with data. Process to the final model may not be interesting for most readers.

4. Thank you for valuable suggestion. We tried two different approaches to establish the IU model of nursing and pharmacy students. One is based on SEM described in this manuscript, others is based on EFA, you suggested in your review comment. Actually, we obtained almost same result in both approach. We adopted the SEM approach in this research because arbitrary choice of variables were needed to build the 4 factor model similar to the original IUS model. So, we decided to use SEM approach to build the model in this research.

5. Thank you for this comment. As described in the manuscript, IUS2000 was originally developed for evaluation of IU of mostly younger Japanese students. Therefore, all subdomains in ISU2000 may not be necessarily suitable for nursing and pharmacy students. This may be concern to use ISU2000 to evaluate IU in nursing and pharmacy students. Because the IU model could be built based on ISU2000, most part of IUS2000 was considered to valid for evaluation off IU of nursing and pharmacy students.
We also add comment based on UNESCO statement (Line 217-218)

6. Thank you for your comment. We used two-way ANOVA in the analysis of Table 5, but this did not reveal in the manuscript. Thank you for pointing that out. We added the comment in the materials and method section (Line 111-112) and footnote of Table 5.

7. Thank you this comment. If any literatures related IU in nursing and pharmacy students would be available, we can make an international comparison. Only reports we found was that future performance as nurse and pharmacist was related to the mind of understanding other cultures when they were students. (reference 7 and 8). We expect that researchers in other countries compare IU of their students with Japanese students.

8-11 Thank you for English correction. We modified as reviewer 1 comments.

12. Added a comment for p value in the footnote of Table 4.

Thank you for all your comments and suggestions. They are really useful to improve our manuscript.

Reviewer 2 Report

The present study highlights important aspects to take into account when providing health science related university education in the future, all while using an exhaustive methodology.

One of the limitations is the small sample size. This could be taken into account to design studies in the future in a broader way, which would allow to associate these characteristics to this type of education. In addition, it would be advisable to compare results after clinical rotations. 

Author Response

Dear Reviewer 2

Thank you for your comments and suggestions.

We added the comment for the limitation caused by small sample size in line 270-271 follow your suggestion. This point is very important for readers.

We also have a future plan to compare the IU before and after clinical rotation of nursing and pharmacy students.

We appreciate this suggestion.